# Hepatitis C Core-Antigen Testing from Dried Blood Spots

**DOI:** 10.3390/v11090830

**Published:** 2019-09-06

**Authors:** Mia J. Biondi, Marjolein van Tilborg, David Smookler, Gregory Heymann, Analiza Aquino, Stephen Perusini, Erin Mandel, Robert A. Kozak, Vera Cherepanov, Matthew Kowgier, Bettina Hansen, Lee W. Goneau, Harry L.A. Janssen, Tony Mazzulli, Gavin Cloherty, Robert J. de Knegt, Jordan J. Feld

**Affiliations:** 1Toronto Centre for Liver Disease, Toronto General Hospital, University Health Network, University of Toronto, Toronto, ON M5G 2C4, Canada (M.J.B.) (M.v.T.) (D.S.) (G.H.) (E.M.) (V.C.) (B.H.) (H.L.A.J.); 2Viral Hepatitis Care Network (VIRCAN) Study Group, Toronto Centre for Liver Disease, Toronto, ON M5G 2C4 Canada; 3Arthur Labatt Family School of Nursing, Western University, London, ON N6A 3K7, Canada; 4Gastroenterology and Hepatology, Erasmus MC University Medical Center Rotterdam, Rotterdam 3015 GD, The Netherlands; 5Mount Sinai Hospital, Toronto, ON M5G 1X5, Canada (A.A.) (T.M.); 6Public Health Ontario Laboratories, Toronto, ON M5G 1M1, Canada (S.P.) (L.W.G.); 7Department of Microbiology, Sunnybrook Health Sciences, Toronto, ON M4N 3M5, Canada; 8Dalla Lana School of Public Health, University of Toronto, Toronto, ON M5T 3M7, Canada; 9Abbott Molecular, Des Plaines, IL 60018, USA; 10Institute of Medical Sciences, University of Toronto, Toronto, ON M5S 1A8, Canada

**Keywords:** Chronic hepatitis C, dried blood spot, core-antigen, diagnosis, widespread screening

## Abstract

In order to expand hepatitis C virus (HCV) screening, a change in the diagnostic paradigm is warranted to improve accessibility and decrease costs, such as utilizing dried blood spot (DBS) collection. In our study, blood from 68 patients with chronic HCV infection was spotted onto DBS cards and stored at the following temperatures for one week: −80 °C, 4 °C, 21 °C, 37 °C, and alternating 37 °C and 4 °C; to assess whether temperature change during transportation would affect sensitivity. Sample was eluted from the DBS cards and tested for HCV antibodies (HCV-Ab) and HCV core antigen (core-Ag). HCV-Abs were detected from 68/68 DBS samples at −80 °C, 4 °C, 21 °C, and 67/68 at 37 °C and alternating 37 °C and 4 °C. Sensitivity of core-Ag was as follows: 94% (−80 °C), 94% (4 °C), 91% (21 °C), 93% (37 °C), and 93% (37 °C/4 °C). Not only did temperature not greatly affect sensitivity, but sensitivities are higher than previously reported, and support the use of this assay as an alternative to HCV RNA. We then completed a head-to-head comparison (*n* = 49) of venous versus capillary samples, and one versus two DBS. No difference in core-Ag sensitivity was observed by sample type, but there was an improvement when using two spots. We conclude that HCV-Abs and core-Ag testing from DBS cards has high diagnostic accuracy and could be considered as an alternative to HCV RNA in certain settings.

## 1. Introduction

Therapeutic developments have revolutionized the treatment of hepatitis C virus (HCV) infection and raised the prospect of global elimination. Despite this progress, HCV continues to have low rates of diagnosis, partially due to complicated diagnostic algorithms. The most widely used approach is the two-assay format that detects HCV antibodies (HCV-Abs), followed by viral RNA to confirm current infection [1]. Although both assays can be done from a single sample, most laboratories require two samples for reflex testing, or a follow-up sample, which has been cited as a barrier in the HCV cascade of care (reviewed in Reference [2]). Alternatively, a measure of active infection could be done as the initial test, however the cost of HCV RNA is often prohibitive. HCV core antigen (core-Ag) is an alternative marker of infection with high specificity, and correlates well with HCV viral load. The clinical sensitivity of the assay is excellent; however, the analytical sensitivity is unreliable below 3–10 fmol/L (RNA viral load of 1000–3000 IU/mL) [3]. Nonetheless, as most individuals with chronic HCV infection have RNA levels above this threshold, core-Ag is an effective and less expensive tool for confirmation of active infection [4].

In many high-income countries, up to half of chronic infections remain undiagnosed and diagnosis rates in low- and middle-income countries are much lower [5,6,7]. The low diagnosis rates are due to the generally asymptomatic nature of the disease, the reliance on risk factor-based testing, and in some settings, limited access to healthcare infrastructure and the high cost of HCV diagnostic tests [8,9]. Among high-risk populations, additional barriers to testing exist. For people who inject drugs, standard venipuncture may be particularly challenging. The need for trained staff to draw venous blood limits peer testing and other approaches that may help engage people who may have experienced stigma in healthcare settings [10]. The requirement to come back for a second test is a major additional barrier, with 30–40% of persons who test HCV antibody positive never returning to confirm viremia, let alone to engage in care and treatment [11,12]. Dried blood spot (DBS) testing has been shown to overcome some of these challenges and is widely used in other diseases [13]. As only a finger-prick is needed, the requirement for venous access is eliminated, allowing those without healthcare training to perform sample collection. Furthermore, subsequent testing can be reflexively completed from the same card.

The HCV core-Ag assay from DBS could provide a less expensive alternative to HCV RNA testing for active infection, however, there have been few studies that have assessed the sensitivity and quantification of core-Ag from DBS, and those that have do not have consistent collection or processing methods. Furthermore, the effects of storage conditions are not known [14]. The aim of this study was to determine whether testing core-Ag from DBS could be used to diagnose chronic HCV infection; and whether storage temperature influences diagnostic sensitivity and the correlation between core-Ag from serum and DBS. In follow-up we determined whether sensitivity differed when using venous versus capillary blood, as well as one versus two dried blood spots. 

## 2. Materials and Methods 

### 2.1. Study Population and Design

Patients with chronic HCV infection visiting the outpatient clinic of the Toronto Centre for Liver Disease (TCLD) (Toronto, Canada) were approached from June 2015 to December 2018. All HCV-positive persons included in the study had a detectable viral load as determined clinically at Public Health Ontario by the Roche Cobas AmpliPrep TaqMan HCV Assay, and were either treatment-naïve or were classified based on their prior treatment history. Clinical and demographic data were collected from the electronic medical record. To assess the specificity of DBS, known HCV-negative controls were included. Staff carrying out the testing of DBS samples were blinded to the HCV status of the tested individuals.

### 2.2. Dried Blood Spots (DBS) and Serum Sample Collection

In order to assess overall sensitivity and the effect of storage conditions, two tubes of venous blood were collected from HCV-infected individuals and negative controls. One tube was used to fill five spots of 50 µL per spot to a Munktell-TFN protein saving DBS cards (Ahlstrom Germany, Bärenstein, Germany). The spotted DBS cards were left to dry overnight at room temperature and were then inserted into an individual plastic bag containing a desiccant pack. Each DBS card was subsequently stored under a different condition for one week to simulate issues that may arise during sample collection and transport: (1) −80 °C (gold standard), (2) 4 °C (refrigeration rather than freezing), (3) 21 °C (storage at room temperature rather than freezing), (4) 37 °C (storage in high temperature environments) and (5) three days of alternating 12 h at 37 °C and 12 h at 4 °C (variable temperatures during transport). Following the experimental storage conditions, the DBS cards were stored at −80 °C until elution. The second tube of venous blood collected was stored at −80 °C for serum core-Ag testing. 

In order to evaluate the effect of capillary blood collection in comparison to venous blood, one tube of venous blood was collected from patients with chronic HCV infection and HCV Ab-negative individuals. This tube was used to spot 75 µL of blood per spot to six spots on two Munktell-TFN protein saving DBS cards (Ahlstrom Germany, Bärenstein, Germany). Additionally, three dried blood spots were filled by capillary blood from a finger-prick sample from the same individual. The spotted DBS cards were left to dry overnight at room temperature and were then inserted in an individual plastic bag containing a desiccant pack. Each DBS card was subsequently stored at −80 °C until elution. 

### 2.3. Dried Blood Spots (DBS) Anti-Hepatitis C Virus (HCV) and Core-Antigen (Ag) Elution and Testing

After storage at −80 °C, the individual spots were cut out and inserted into a microtube containing 0.8 mL of 0.25% TX-100 phosphate-buffered saline (PBS) for elution. 400 µL was used for testing; or in the optimized protocol 0.8 mL for one, two, or three spots (three for venous only). Microtubes were gently shaken and incubated at room temperature for 90 min. Spots were then removed, and the elution product was stored at −80 °C until diagnostic testing. HCV-Abs were tested using the Architect i4000 SR (Abbott, Chicago, Illinois) with a lower limit of detection (LLOD) of 1.00 S/CO and core-Ag was tested by the HCV ARCHITECT core antigen assay (Abbott, Chicago, Illinois) with a LLOD of 3 fmol/L. In this assay values of 3–10 fmol/L are diagnostically positive (grey-zone), but the lower limit of quantification is 10 fmol/L. Thus, only samples that were determined to be greater than 10 fmol/L were correlated to the patient’s viral load. However, all samples greater than 3 fmol/L were included for determination of sensitivity. 

### 2.4. Statistical Analysis

Data are presented as medians with interquartile range (IQR) or means with standard deviations (SD). Sensitivity was calculated to determine diagnostic accuracy. Quantitative data were compared using a paired t-test. Correlation between quantitative samples was assessed using Pearson’s correlation coefficient after log transformation. Analysis and graphing were completed using Prism GraphPad (version 7). 

### 2.5. Ethics

The study protocol was conducted in accordance with the Declaration of Helsinki and Good Clinical Practice guidelines and was originally approved on the 16th of December 2014 by the University Health Network Ethics Review Board (UHN REB 14-8383).

## 3. Results

### 3.1. Patient Characteristics

A total of 117 patients with chronic HCV infection, and 15 HCV-Ab negative patients were included. Within the 15 negative controls, five were healthy individuals, and 10 were infected with hepatitis B virus (HBV). Patient characteristics are summarized in Appendix A. The majority of the cohort was male (59.0%), and non-cirrhotic (70.9%). The most frequently identified genotype was genotype (GT) 1 (65.8%), although other genotypes were also reported. With respect to treatment status, 89 (76.1%) patients were treatment-naïve, 24 (20.5%) were prior non-responders or had relapsed, and 4 (3.4%) discontinued therapy due to adverse events.

### 3.2. Detection of Core-Antigen (Ag) Is Unaffected by Changes in Storage Temperature 

Testing algorithms for HCV include screening samples for the presence of HCV Abs, followed by a confirmatory assay to establish active infection. In our study, HCV-Abs were detected from DBS in all HCV-infected patients in all conditions, except for one patient in whom HCV-Abs were not detected from a DBS card stored at 37 °C and at 37 °C/4 °C (Table 1). Next, we examined whether temperature affected core-Ag detection. Storage at −80 °C and 4 °C had the highest sensitivity at 94.1% (95% CI 88.5–99.7%), and overall differences in sensitivity between conditions were minimal ranging from 91.2–94.1%. However, in 4 samples there were conditions in which quantitative results were in, or slightly above the grey-zone (3–10 fmol/L) for certain conditions, but below the LLOD for others. In only one of these scenarios was there a pattern with hypothesized degradation based on temperature, i.e., positive in −80 °C, 4 °C, 25 °C, and negative at 37 °C and 37/4 °C. When analysis was performed using the quantification cutoff value of ≥10 fmol/L, detection ranged from 80.9–85.3%. All HCV-negative controls tested negative for core-Ag from DBS under all storage conditions. Sensitivity for all five storage conditions from DBS is shown in Table 1.

### 3.3. Quantitative Analysis of Core-Antigen (Ag) from Dried Blood Spots (DBS) 

Quantification of HCV viral load may have utility in treatment decisions. Therefore, we investigated the correlation between serum core-Ag levels and HCV RNA. As shown in Appendix A, we determined that core-Ag concentration in serum significantly correlated with HCV-RNA concentration in serum (r = 0.76 (95%CI 0.63–0.84), *p* < 0.0001). Based on the serum viral RNA, one sample was predicted to be below the LLOD based on established cut-offs for serum core-Ag (1000–3000 IU/mL). Additionally, concordance between DBS duplicates for detection of HCV-Abs and detection of core-Ag was calculated and ranged from 96–100% depending on storage condition (Appendix A). Agreement when using a core-Ag quantitative cut-off of 10 fmol/L is also shown (Appendix A).

Interestingly, mean core-Ag concentration in serum was 1.7–1.9 log higher than core-Ag concentration from DBS irrespective of the storage condition (*p* < 0.0001) (Figure 1). There was also a statistically significant difference between mean values of samples stored at −80 °C in comparison to samples stored at 37 °C and 37/4 °C (*p* < 0.05). The correlations between core-Ag concentration in serum and DBS for the different conditions varied and demonstrated a statistically significant moderate to strong positive correlation. As a representative example, a correlation graph for −80 °C including all samples is shown in Figure 2A, r = 0.71. R-values were as follows for the various conditions: r = 0.76 (−80 °C, Figure 2B), 0.71 (4 °C, Figure 2C), 0.73 (21 °C, Figure 2D), 0.45 (37 °C, Figure 2E), 0.60 (37 °C/4 °C, Figure 2F). 

### 3.4. Core Antigen (Ag) Sensitivity Is Increased by Dried Blood Spots (DBS) Number

Several groups have assessed the sensitivity of core-Ag from DBS using different sample collection methods and sample processing conditions with variable sensitivity reported [15,16,17]. We investigated whether the collection method (venous blood vs. finger-prick blood) and number of spots eluted (one vs. two) from the DBS card affected assay sensitivity. As we demonstrated that storage temperature does not significantly affect sensitivity, for this analysis all cards were stored at −80 °C until elution. Sensitivity for both sample types and number of dried blood spots are shown in Appendix A. For diagnostic sensitivity, i.e., >3 fmol/L results were as follows: for both one venous sample spot and one finger-prick spot sensitivity was 91.8% (95% CI 84.2–99.5%); for two venous sample spots and two finger-prick spots sensitivity was 93.9% (95% CI 87.2–100%). There were 3 samples for which core-Ag was not detected across all four conditions. The first sample was below the limit of detection of 3000 IU/mL with a viral load of 1710 IU/mL. The other two samples had serum HCV RNA levels of 13,300 and 47,200 IU/mL; which were confirmed by chart review as chronic infections (despite low viral loads).

In addition, in two other samples, results went from undetectable (<3 fmol/L) to detectable (>3 fmol/L) when two spots were used instead of one. The first of the two was above the LLOD in one and two DBS blotted by finger-prick, however, when the sample was collected by venipuncture, only the sample eluted from two spots was above the LLOD. In the second patient, the opposite was true, where samples from both one and two DBS from venous blood were above the LLOD, but two spots were necessary for positive detection from the finger-prick sample.

In those that were undetectable across all four conditions, no change in sensitivity was observed when a third venous spot was used (data not shown). In both capillary and venous samples, quantitative data for two dried blood spots was approximately two-fold that of one DBS: venous 1.95-fold (SD +/− 0.35); finger-prick 1.97-fold (SD +/− 0.44). All 10 known HCV Ab negative samples were correctly identified as negative.

There was a strong correlation in the quantitative analysis between venous and finger-prick samples. As is shown in Figure 3, this was observed when comparing one spot of venous blood to finger-prick r = 0.97 (95%CI 0.94–0.98), and when comparing two spots of venous to finger-prick whole blood r = 0.98 (95%CI 0.96–0.99). Next, we compared one versus two DBS from finger-prick samples, which had been stored at −80 °C as these are the more clinically relevant sample type. Our findings demonstrate that there is a strong positive correlation between two DBS samples blotted from finger-prick in comparison to HCV RNA from serum, r = 0.9 (Figure 3C), as there is for a single DBS (data not shown). 

With respect to ability to provide quantitative data which can be correlated to viral load (i.e., >10 fmol/L), sensitivity for one venous sample spot was 87.8% (95% CI 79.0–97.0%); one finger-prick spot, 81.6% (95% CI 71.4–92.6%); two venous sample spots, 87.8% (95% CI 79.0–97.0%); and two finger-prick spots, 85.7% (95% CI 76.4–95.6%) (Appendix A). The addition of an extra spot resulted in two samples collected by finger-prick to increase from between 3 fmol/L-10 fmol/L to >10 fmol/L. This same observation did not occur with venous samples; however, there were few patient samples with viral loads near the threshold for detection by core-Ag.

## 4. Discussion

Identifying HCV cases requires a two-tiered testing algorithm involving detection of HCV-Abs and HCV-RNA (or another marker of active infection such as core-Ag). Testing from DBS has been recommended as a promising alternative to testing from plasma or serum in recent guidelines from the WHO. The sensitivity and specificity of HCV-Ab testing from DBS is 98% and 99%, respectively, while HCV RNA testing from DBS has a sensitivity of 96% and a specificity of 98% [18]. However, to date the use of HCV core-Ag from DBS as a diagnostic assay for HCV infection has not been recommended due to the limited data assessing the sensitivity and utility in real-world settings. Our study lends support for this platform as an option for diagnosing chronic HCV infection, and in fact shows increase sensitivity compared to other studies (Table 2).

The sensitivity of antibody testing demonstrated here is in agreement with results published previously [16,19], and none of our HCV-negative controls tested positive for either HCV-Ab or core-Ag from DBS. Although the number of HCV-negative controls was very limited, the specificity of HCV-Abs and core-Ag from DBS has been reported elsewhere [16,17]. There was good concordance between duplicates for antibody and antigen from a diagnosis perspective, demonstrating that duplicate testing is not necessary.

Our study focused on individuals with detectable viral load in a clinical setting for the purposes of diagnosing chronic infection, as case-finding remains a major challenge in the elimination of HCV globally, hindered by collection difficulties and high costs of testing. Other studies have also evaluated the use of core-Ag from DBS to confirm undetectable RNA. Soulier et al. assessed 26 patients with known resolved infection as defined by the authors as either acutely clearing (*n* = 13) or SVR (*n* = 13), and confirmed that core-Ag was undetectable in all cases (specificity of 100%) [16]. Moreover, a recent study demonstrated that HIV or hepatitis B virus co-infection did not affect the sensitivity of core-Ag testing from DBS [15].

In our hands, we observed that serum had a lower limit of detection than DBS. A similar phenomenon was observed with HCV RNA, relating primarily to sample volume [20]. There was a correlation between quantitative core-Ag measurements between serum and DBS, particularly for samples above 3 fmol/L, which is in agreement with what has been reported elsewhere [16]. We also found minimal differences in sensitivity and quantitative correlation between storage temperatures, in line with others that compared storage at −80 °C to 25 °C [16]. However, for low-quantity samples (i.e., 3–10 fmol/L), temperatures including 37 °C and changes in temperature (37/4 °C) may slightly affect diagnostic sensitivity. Our environmental conditions data support the use of core-Ag from DBS in remote regions where the sample could undergo fluctuations in storage temperature during collection and transit.

There is notable variability in the reported sensitivity of core-Ag from DBS across studies (Table 2). All studies included samples with diverse genotypes, and no genotype-specific changes in sensitivity have been observed [15,16]. There are several possible reasons for the differences in sensitivity between studies. Although each study used the Abbott core-Ag detection system, differences in collection methods, as well as sample processing (number of spots, elution time) have not been standardized for core-Ag processing from DBS, possibly leading to large reported differences in sensitivity ranging from 64–83% [16,20]. We showed that the sample type (venous blood vs. finger-prick) used for core-Ag from DBS did not affect the results. Importantly, we also showed that doubling the number of spots doubles the quantitative core-Ag results, which is relevant for samples near the limit of detection. In our study, one sample was below the cut-off of 3000 IU/mL recommended for core-Ag testing from serum. Thus, if we were to use the previously established HCV RNA cut-off for the core antigen assay, our sensitivity would be 96.9% (95% CI 92.0–100%). However, in order to reliably capture sensitivity for samples less than 1.0 × 10^5^ IU/mL in the context of DBS, larger studies focusing on these viral loads will need to be completed. These data will be useful to continue to standardize processing guidelines for the microbiology diagnostics community [21].

In order to work towards the global elimination of HCV, different testing methodologies will be required. The optimal diagnostic approaches vary depending on disease prevalence, testing setting and the availability of testing options including point-of-care antibody or HCV RNA testing, DBS collection ability, and access to facilities for antibody, RNA, and core-Ag high-throughput sample processing [19]. Potential testing algorithms that incorporate DBS are outlined in Figure 4. The cost of both testing and treatment may also influence the modality chosen in specific settings. With respect to DBS, a British study looking at direct costs of HCV antibody testing from DBS versus venous sampling showed that the cost per positive test was three times less using DBS; and other studies demonstrate that HCV RNA DBS testing increases access to services in non-clinical settings [22,23,24,25]. Therefore, the use of DBS for antibody screening with reflex RNA testing should be considered in areas where sample collection and transportation are difficult (Figure 4A). Alternatively, to reduce HCV RNA testing, the use of DBS for initial HCV-Ab screening, with a core-Ag reflex test, and a follow-up RNA test (for patients who are HCV-Ab positive, but core-Ag negative) may also be considered in resource-rich settings (Figure 4B). Although markedly improved in compared to other studies, the qualitative sensitivity of core-Ag testing was still only in 91–94% range, raising concern of false-negative results if confirmatory HCV RNA testing is not added for Ab-positive/core-Ag-negative samples (Figure 4C). Whether it would be more cost-effective to forego confirmatory RNA testing would depend primarily on the costs of the different assays but also on the frequency of spontaneous HCV clearance. Moreover, it must be considered in the context of patients with very-low viral loads, as it has recently been shown that core-Ag may not be detected [26].

In settings such as Sub-Saharan Africa, where higher rates of spontaneous HCV clearance are reported, likely due to the high prevalence of past or current hepatitis B co-infection, more confirmatory HCV RNA tests would be required [27,28,29], which is costly. Thus, as core-Ag is less expensive than HCV RNA, it may be reasonable to first look for active infection by this method and send all negative samples for RNA testing.

Finally, direct HCV core-Ag testing without initial Ab testing from DBS may be an appropriate initial screening method in high prevalence populations such as in people with ongoing injection drug use, and a less expensive alternative to HCV RNA testing (Figure 4D). However, the amount saved by eliminating the HCV-Abs test must be weighed against the knowledge gained by knowing an individual is antibody positive, and the potential for education or contact tracing. It should be noted, that although predominately in early acute infection, the possibility for a patient to be HCV-Ab negative, but RNA positive does exist. Nonetheless, a major advantage of all algorithms (Figure 4A–D) is that additional testing can be done from the same card, reducing lost to follow-up rates before establishing chronicity. Although additional testing could include genotyping, the relevance of this test considering the cost, may be less important in the era of multiple pan-genotypic treatment options. 

This study has important limitations. Specifically, few HCV RNA negative samples were included and no samples post-SVR or spontaneous HCV RNA clearance were included, however, the specificity of core-Ag testing has been well validated in many studies, as described above. Considering the small number of samples with HCV RNA levels less than 10,000 IU/mL, it may be difficult to determine whether the number of spots has an effect on sensitivity. However, our quantitative data clearly demonstrate that doubling the number of spots eluted increases the amount of core-Ag detected.

In conclusion, we have demonstrated that HCV-Ab and reflex core-Ag testing from DBS has a higher sensitivity for diagnosis (~94%) than previously reported. Quantitative DBS core-Ag level correlates adequately to core-Ag levels in serum particularly given that the quantitative results are of limited clinical value. We also show that HCV-Ab and core-Ag results from DBS samples are not affected by storage temperature, as evidenced by only small differences in sensitivity, important in situations where sample shipment can be expensive, and samples may be in transit for several days. Finally, we demonstrate that while there is no difference between sample collection methods. There is a linear relationship between the spot number and the quantitative HCV core-Ag result, which has the potential to increase sensitivity of the assay if samples are slightly below the 3 fmol/L cut-off. Our data would support using LLOD (3 fmol/L) rather than the lower limit of quantification (10 fmol/L) when using core-Ag off DBS to improve overall sensitivity.

Our results are particularly relevant for screening in difficult-to-reach populations, where eliminating the need for a venous sample could increase peer screening; and in rural and remote populations, where there may be a lack of trained personnel and samples may have to undergo various temperature changes in transit. Finally, the use of core-Ag from DBS may reduce the need for HCV RNA testing, with the potential for cost-saving.

## Figures and Tables

**Figure 1 viruses-11-00830-f001:**
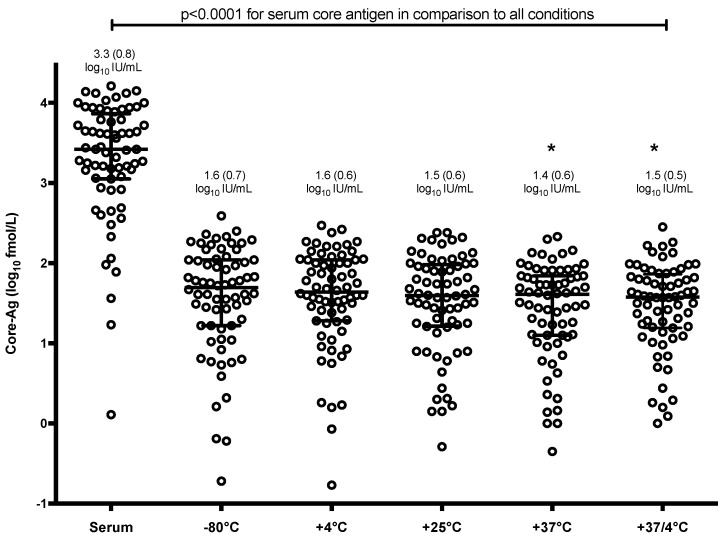
Distribution of core-Ag concentration by dried blood spots (DBS) storage condition as compared to serum. Bars represent mean and standard deviation. Environmental conditions with an astrix (*) have a mean statistically significant difference from −80 °C (*p* < 0.05).

**Figure 2 viruses-11-00830-f002:**
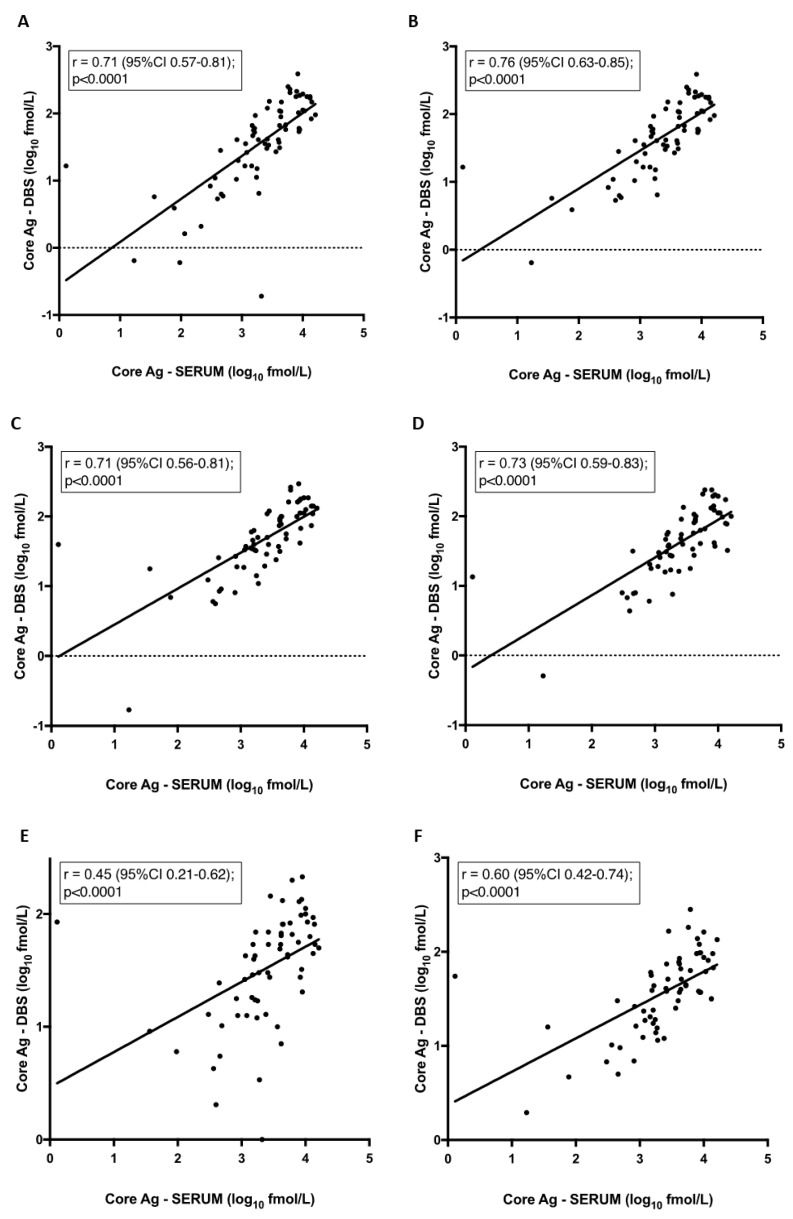
Correlation between core-antigen (Ag) titre from dried blood spots (DBS) and serum per condition. Pearson correlation: (**A**) Samples stored at −80 °C. Samples excluded if core-Ag concentration was less than 3 fmol/L stored at (**B**) −80 °C, (**C**) 4 °C, (**D**) 21 °C, (**E**) 37 °C, (**F**) 37 °C/4 °C.

**Figure 3 viruses-11-00830-f003:**
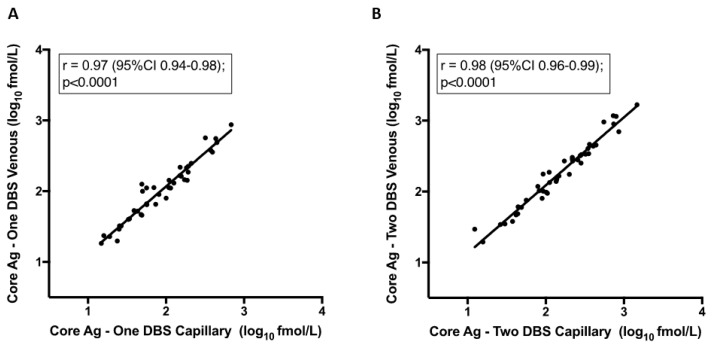
Correlation between core-antigen (Ag) titre from dried blood spots (DBS) from venous and capillary samples and one vs. two spots. Pearson correlation: (**A**) Core-Ag measurement from one DBS from venous whole blood to one DBS from capillary whole blood finger-prick, (**B**) Core-Ag measurement from two DBS from venous whole blood to two DBS from capillary whole blood finger-prick, (**C**) Core-Ag measurement from two DBS from capillary whole blood finger-prick as compared to RNA viral load measurement in serum.

**Figure 4 viruses-11-00830-f004:**
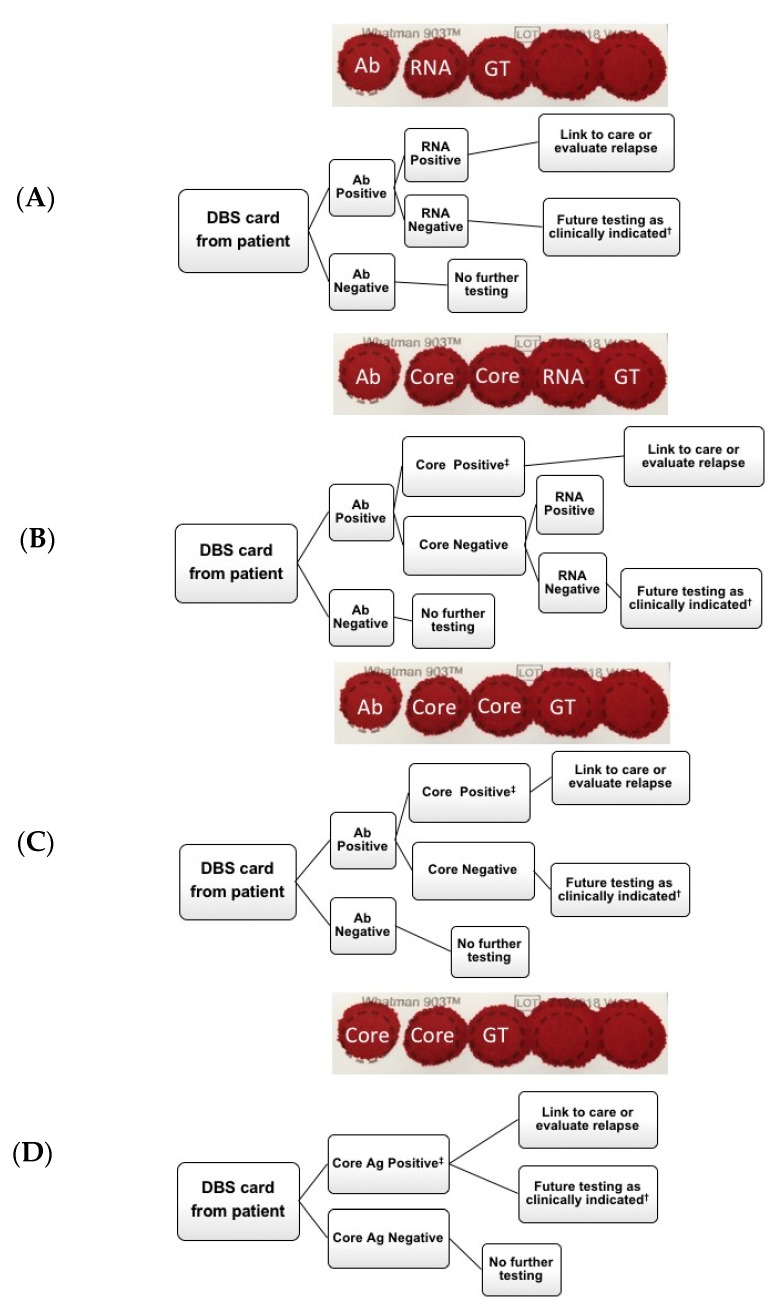
Potential diagnostic algorithms for hepatitis C virus (HCV) testing. (**A**) HCV antibody testing from DBS with subsequent RNA testing, (**B**) HCV antibody testing from dried blood spots (DBS) with subsequent core-Ag testing. If core-antigen (Ag) negative, subsequent RNA testing, (**C**) HCV antibody testing from DBS with subsequent core-Ag testing, (**D**) HCV core-Ag testing from DBS card. ^†^ Retest as clinically indicated if suspected seroconversion or ongoing exposure. ^‡^ Testing ends as a result of positive HCV core antigen. Ab = antibody, Core = core antigen, GT = genotype.

**Table 1 viruses-11-00830-t001:** Sensitivity and specificity of hepatitis C Virus (HCV) antibody and antigen on dried blood spots (DBS) by storage condition.

Storage Condition	HCV Antibody	HCV Core Antigen	
	Sensitivity (95% CI) *n* = 68	Sensitivity >3 fmol/L (95% CI) *n* = 68	Sensitivity >10 fmol/L (95% CI) *n* = 68
**−80 °C**	100% (100–100) (68/68)	94.1% (88.5–99.7) (64/68)	85.3% (76.4–94.2) (58/68)
**+4 °C**	100% (100–100) (68/68)	94.1% (88.5–99.7) (64/68)	85.3% (76.4–94.2) (58/68)
**+21 °C**	100% (100–100) (68/68)	91.2% (84.3–98.1) (62/68)	80.9% (70.8–91.0) (55/68)
**+37 °C**	98.6% (96.6–100) (67/68)	92.7% (86.4–98.9) (63/68)	80.9% (70.8–91.0) (55/68)
**+37 °C/+4 °C**	98.6% (96.6–100) (67/68)	92.7% (86.4–98.9) (63/68)	85.3% (76.4–94.2) (58/68)

**Table 2 viruses-11-00830-t002:** Summary of sensitivity and specificity of hepatitis C virus (HCV) diagnostic options.

Test	Biondi et al.		Mohamed et al. [15]		Lamoury et al. [17]		Soulier et al. [16]		WHO Meta-Analysis [19]	
	Sensitivity ^†^	Specificity	Sensitivity ^†^	Specificity ^†^	Sensitivity ^†^	Specificity ^†^	Sensitivity ^†^	Specificity ^†^	Sensitivity ^†^	Specificity ^†^
HCV-Ab									98.0% (98.0–100)	100% (100–100)
HCV RNA									100% (100–100)	99.9% (99.5–100)
HCV-Ab DBS	100% (96.4–100)						99.1% (97.4–99.8)	98.2% (94.9–99.6)	98.0% (94.0–99.0)	99.0% (97.0–100)
HCV RNA DBS ^‡^							98.1% (95.9–99.1)	100% (97.8–100)	96.0% (93.4–97.6)	97.7% (94.7–99.0)
Core-Ag DBS (−80 °C)	94.1% (88.5–99.7)	15/15	76.7%	97.3%	82.9% (74–90)	96.1% (78–100)	64.1% (58.5–69.3)	100% (97.8–100)		
HCV-Core-Ag DBS (21–24 °C)	91.2% (84.3–98.1)	5/5					64.1% (58.5–69.3)	100% (97.8–100)		

^†^ 95% CI. ^‡^ Combined analysis of venous and capillary blood.

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
