# Peer review of "Hepatitis C Core-Antigen Testing from Dried Blood Spots"

_viruses, 2019, doi:10.3390/v11090830_

Round 1

Reviewer 1 Report

The aim of this study was to compare the sensitivity and specificity of HCV antibody and core antigen testing from a DBS subjected to different storage conditions, including conditions which might mimic those encountered in low income countries.  The manuscript was well written, the introduction relevant to the topic and the figures complemented the written text.  The conclusion, that testing from DBS cards could be considered in certain settings is justified, but I believe unlikely to be accepted until studies with larger samples numbers are completed.

There are several points which should be addressed;

Line 28-"Cards were eluted"-the intention is clear but the wording is nonsense and typical of an undergraduate text.

Line 29-it would be helpful in the abstract to confirm that HCV-Abs were detected from all DBS in all conditions other than a single samples in the storage conditions which included 370C.

Line 88-what was the volume of the spots in these samples.

Table 1-I would prefer if the authors included the primary data viz 68/68 etc.

Figure 2 legend-I was confused at the term "All samples stored at -800C" I think that "All samples" would suffice.

Lines 198-211-I read this section several times in an effort to understand the intended meaning, but as casual readers are unlikely to do so, perhaps the authors could attempt to clarify things. Line 202-"The first"-I assume this means the first sample, but it would help if this was clarified and the text in lines 205/206 was most confusing.  Would a small table help here?

Line 260-please include the references to the other studies.

Line 275-this is a question of semantics, but I understand "The limit of detection of core-Ag assay from DBS was lower" to mean that this assay was more sensitive, but I doubt if this is the intending meaning.

Figure 4. The proposed algorithms are potentially useful, but the authors should remind the reader of the potential for HCV Ab-negative, HCV RNA positive samples.

Author Response

Response to Reviewer 1

Thank you kindly for the reviews of our manuscript.

Below is a point-by-point response to your comments including line references in the revised version.

Point 1: Line 28-"Cards were eluted"-the intention is clear but the wording is nonsense and typical of an undergraduate text.

Response 1: Changed to: “Sample was eluted from the DBS cards…” Now line 29.

Point 2: Line 29-it would be helpful in the abstract to confirm that HCV-Abs were detected from all DBS in all conditions other than a single samples in the storage conditions which included 370C.

Response 2: Excellent point, much clearer. Changed to: “HCV-Abs were detected from 68/68 DBS at -80oC, +4oC, +21oC, and 67/68 at +37oC and alternating +37oC and +4oC, for an overall sensitivity of 99.4%.”. Now lines 30-31.

Point 3: Line 88-what was the volume of the spots in these samples.

Response 3: 50 µL. Important point, as in the second set of experiments to optimize, we used 75 µL. Added on line 94.

Point 4: Table 1-I would prefer if the authors included the primary data viz 68/68 etc.

Response 4: Thank you. I am not sure of the aesthetics of adding such data to the table. However, it is currently added and will leave to the copy editor to address formatting.

Point 5: Figure 2 legend-I was confused at the term "All samples stored at -800C" I think that "All samples" would suffice.

Response 5: There may be a misunderstanding of what Figure 2A is demonstrating. No samples are excluded in A, while those less than 3fmol/L are excluded in B-F. I agree however this may be confusing, so removed the word “all”. Now reads as A) Samples stored at -80 ºC. Samples excluded if core-Ag concentration was less than 3 fmol/L stored at B) -80 ºC, C) +4 ºC, D) +21ºC, E) +37ºC, F) +37 ºC/+4 ºC. Lines 206-208.

Point 6: Lines 198-211-I read this section several times in an effort to understand the intended meaning, but as casual readers are unlikely to do so, perhaps the authors could attempt to clarify things. Line 202-"The first"-I assume this means the first sample, but it would help if this was clarified and the text in lines 205/206 was most confusing.  Would a small table help here?

Response 6: Thank you. Revised to make more concise. Lines 223-228.

Point 7: Line 260-please include the references to the other studies.

Response 7: Reference to Table 2, such that the reader is pointed to the differences, and the reference is available. Line 312.

Point 8: Line 275-this is a question of semantics, but I understand "The limit of detection of core-Ag assay from DBS was lower" to mean that this assay was more sensitive, but I doubt if this is the intending meaning.

Response 8: Agreed. Changed to: “In our hands, we observed that serum had a lower limit of detection than DBS. A similar phenomenon with HCV RNA, relating primarily to sample volume [20].” Lines 327-328.

Point 9: Figure 4. The proposed algorithms are potentially useful, but the authors should remind the reader of the potential for HCV Ab-negative, HCV RNA positive samples.

Response 9: Agree. Please see lines 421-423.

Reviewer 2 Report

In this paper, Biondi et al. assessed the sensitivity of HCV core Ag testing from DBS, as a more affordable alternative than standard of qRT-PCR. The manuscript is well written and experiments conducted support the conclusions of the paper.

One main observation from the paper, is the much lower level of core Ag recovered from DBS compared to from venous blood sample. This is of particular relevance, as this will negatively affect the sensitivity of the core Ag assay from DBS, and therefore could prevent more subjects with low viral load to early access to care. The authors should discuss the potential reasons for this observation, and the likely need to optimize/standardize the processing steps to minimize this observation (provided this is not only due to volume).

The manuscript should be proof-read for typos, such as on page 2 line 70 (sentence unclear) or page 12 line 383 (; iso ,).

Author Response

Response to Reviewer 2

Point 1: One main observation from the paper, is the much lower level of core Ag recovered from DBS compared to from venous blood sample. This is of particular relevance, as this will negatively affect the sensitivity of the core Ag assay from DBS, and therefore could prevent more subjects with low viral load to early access to care. The authors should discuss the potential reasons for this observation, and the likely need to optimize/standardize the processing steps to minimize this observation (provided this is not only due to volume).

Response 1: Thank you for this comment. We do hypothesize that in fact this observation is related to volume. Please see line 327-328. We agree that using DBS core-Ag this may affect early access to care for those with low viral loads, which is why we have discussed various algorithms for testing. We agree that these algorithms have specific uses depending on region and prevalence (lines 360-427). For example, Figure 4B would be the ideal in resource-rich settings that are using DBS collection for difficult to reach populations either related to venous access or unable to go to a lab. We are not suggesting substandard testing in resource-limited regions, rather suggesting core-Ag DBS be used where there is no possibility to send a venous sample to a central lab, but where the central lab could utilize a less cost-prohibitive approach such as core-Ag as compared to RNA, and obtain the sample from DBS (as RNA is approximately 5 times the cost).

Point 2: The manuscript should be proof-read for typos, such as on page 2 line 70 (sentence unclear).

Response 2:

Corrected, line 76.

or page 12 line 383 (; iso ,).

Agree, corrected line 441-443.  

The rest of the manuscript has been proof-read, and changes have been tracked.